# Prevalence and Associated Factors of Physical Inactivity Among Malaysian Adults

**DOI:** 10.3390/healthcare13222978

**Published:** 2025-11-19

**Authors:** Kuang Kuay Lim, Hamizatul Akmal Abd Hamid, Muhammad Fadhli Mohd Yusof, Azli Baharudin Shaharuddin, Tuan Mohd Amin Tuan Lah, Ying Ying Chan, Khairulaizat Mahdin, Norliza Shamsudin, Halizah Mat Rifin, Vanitha Subramaniam

**Affiliations:** 1Institute for Medical Research, National Institutes of Health, Ministry of Health, Setia Alam 40170, Malaysia; 2Institute for Public Health, National Institutes of Health, Ministry of Health, Setia Alam 40170, Malaysia; hamizatul_ah@moh.gov.my (H.A.A.H.); fadhli_my@moh.gov.my (M.F.M.Y.); ps_azlibaharudin@moh.gov.my (A.B.S.); tuanmohdamin@moh.gov.my (T.M.A.T.L.); chan.yy@moh.gov.my (Y.Y.C.); khairulaizat@moh.gov.my (K.M.); norlizas@moh.gov.my (N.S.); halizah.matrifin@moh.gov.my (H.M.R.); 3Health Education Division, Community Communications Section, Ministry of Health, Putrajaya 62590, Malaysia; svanitha@moh.gov.my

**Keywords:** physical inactivity, adults, GPAQ, NHMS, Malaysia

## Abstract

**Background**: Regular physical activity benefits people of all ages by enhancing both mental and physical health, whereas inactivity raises the risk of noncommunicable diseases (NCDs) like cardiovascular diseases, cancer, and diabetes, which can result in premature death. The objective of this cross-sectional study was to determine the prevalence and associated factors of physical inactivity among adults in Malaysia. **Methods:** Data on physical inactivity were extracted from the 2023 National Health and Morbidity Survey (NHMS), conducted from July to September 2023 among adults aged 18 and above across all states of Malaysia. Physical activity was assessed using the Global Physical Activity Questionnaire, with individuals classified as inactive if they did not achieve at least 600 metabolic equivalents of task (MET) minutes per week. Logistic regression was used to identify factors associated with physical inactivity. **Results:** A total of 10,858 out of 13,616 respondents participated in the study, resulting in a response rate of 79.7%. The overall rate of physical inactivity was 29.9%. The multivariable logistic regression analysis showed that physical inactivity was significantly higher among urban residents (aOR 1.42; 95% CI: 1.16, 1.75), individuals who were single/divorced/widowed (aOR 1.16; 95% CI: 1.01, 1.33), those not working (aOR 1.61; 95% CI: 1.24, 2.09), and those with sedentary time exceeding four hours per day (aOR 1.46; 95% CI: 1.20, 1.76). Inactivity was also more likely among individuals with diabetes (aOR 1.17; 95% CI: 1.00, 1.37) or disabilities (aOR 1.67; 95% CI: 1.39, 2.02). **Conclusions:** About one-third of Malaysian adults are physically inactive, with higher rates among urban residents, single/divorced/widowed, those not working, those with extended sedentary time, diabetes, or disabilities. Targeted interventions are needed to encourage behaviour change, foster active urban design, and strengthen policies that support active lifestyles.

## 1. Introduction

Physical inactivity is a major contributor to a range of non-communicable diseases (NCDs), including cardiovascular diseases, cancer, and diabetes, all of which significantly impact public health. It is generally characterised as the inability to participate in sufficient physical activity to uphold optimal health, often seen in sedentary behaviours like extended sitting and minimal moving [1]. The negative health consequences of physical inactivity are well-established, and it is recognised as a major global public health concern [2]. According to the World Health Organization (WHO) [1], insufficient physical activity is one of the leading risk factors for global mortality, contributing substantially to the overall disease burden. The WHO recommends that adults engage in at least 150–300 min of moderate intensity or 75–150 min of vigorous intensity physical activity per week to reduce health risks. Despite increased awareness, physical inactivity remains widespread, posing ongoing challenges to public health systems worldwide [3,4].

Globally, the prevalence of physical inactivity continues to rise, with the WHO estimating that more than one in four adults worldwide do not meet the recommended physical activity levels [1]. This trend significantly contributes to the growing burden of chronic diseases, with approximately 1.8 billion people not meeting the recommended levels of physical activity in 2022 [5]. In high-income countries such as the United States, the United Kingdom, and Australia, inactivity rates are notably high, with around 36.8% of adults not meeting the recommended activity levels [6]. In contrast, low-income countries like India, Nigeria, and Bangladesh report much lower inactivity rates, at approximately 16.2%. Meanwhile, in middle-income nations such as Brazil and South Africa, inactivity rates fall between those of high and low-income countries [5]. These global patterns demonstrate how socio-economic position, urbanisation, technological improvements, and availability of safe areas for physical activity play an important role in determining physical activity levels across different countries and income groups [6].

In Malaysia, physical inactivity remains a significant public health concern, contributing to the increasing prevalence of obesity and NCDs. The National Health and Morbidity Survey (NHMS) 2015 reported that approximately 34% of Malaysian adults were physically inactive [7]. In the subsequent NHMS 2019 survey, the prevalence of physical inactivity among Malaysian adults decreased significantly to 25.1% [8]. One possible reason for the decrease in physical inactivity is the introduction of the Malaysia National Strategic Plan for Active Living (NASPAL) in 2017, which promotes active lifestyles, reduces sedentary behaviour, and encourages active commuting among adults [8]. Despite these improvements, physical inactivity continues to pose a substantial public health challenge, driven by factors such as rapid urbanisation, reliance on motorised transportation, increased screen time, and limited access to safe recreational spaces, particularly in urban areas [9,10]. Therefore, this study aimed to determine the prevalence and associated factors of physical inactivity among adults in Malaysia.

## 2. Methodology

### 2.1. Research Design and Type

This cross-sectional population-based survey, using the Malaysia National Health and Morbidity Survey (NHMS) 2023, was designed to provide a nationally representative sample of adults aged 18 years and above in Malaysia.

### 2.2. Population and Sample

This survey collected data on various aspects of the country’s disease burden and various health-related issues. A two-stage stratified random sampling method was employed with a 95% confidence interval and 0.01–0.07 margin of error, adjusted for design effect (2.0) and 35% non-response to select representative samples of adults from across the country. The sampling frame was derived from the 2020 National Population and Housing Census conducted by the Department of Statistics. A total of 5988 living quarters (LQs) were selected from 499 Enumeration Blocks (EBs). The sample included adults aged 18 and above who had resided in the selected LQs for at least two weeks, excluding institutionalised populations. All households and eligible respondents within each selected LQ were included in the survey. A detailed description of the survey methodology is available in the NHMS 2023 technical report [11]. In this study, data from 10,858 Malaysian adults aged 18 and above were analysed.

### 2.3. Instruments and Techniques

Data were collected using a validated bilingual (English and Malay) structured questionnaire, administered face-to-face via mobile devices. The questionnaire assessed the factors described below.

#### 2.3.1. Physical Activity Assessment

Physical activity levels were assessed using the short version of the Global Physical Activity Questionnaire (GPAQ), developed by the World Health Organization (WHO) [12]. The GPAQ is a validated and reliable tool for assessing physical activity in adult populations. It includes questions on the frequency and duration of work, transport, and leisure over the past week, as well as sedentary behaviour. Individuals accumulating fewer than 600 metabolic equivalent of task (MET) minutes per week from all domains of physical activity were classified as physically inactive, while those with 600 or more MET-minutes/week were classified as physically active [13]. Sedentary behaviour was classified into three categories: ≤2 h/day (low), >2–4 h (moderate), and >4 h (high).

#### 2.3.2. Socio-Demographic Characteristics

The socio-demographic characteristics of the respondents included location (urban, rural), sex (male, female), age (18–39 years, 40–49 years, 50–59 years, 60 years and above), ethnicity (Malay, Chinese, Indian, Other Bumiputera, Others), education level (none, primary, secondary, tertiary), marital status (single/divorced/widowed, married), occupation (government, private, self-employed, not working), and household income (Q1—poorest 20%, Q2, Q3, Q4, Q5—richest 20%). The age group categorisation in this study differs from previous NHMS surveys due to analytical requirements

#### 2.3.3. Lifestyle-Related Variables

The lifestyle behaviours assessed in this study included body mass index (BMI), smoking, alcohol consumption, and sedentary time. The body weight and height of respondents were measured using a TANITA Digital Weighing Scale HD 319 (TANITA Corp., Tokyo, Japan) and a SECA Portable Stadiometer 213 (SECA GmbH & Co. KG, Hamburg, Germany), respectively. Body mass index (BMI) was calculated as weight in kilograms divided by the square of height in meters (kg/m^2^) and was categorized according to the WHO classification; underweight (<18.5 kg/m^2^), normal (18.5–24.9 kg/m^2^), overweight (25.0–29.9 kg/m^2^), and obese (≥30.0 kg/m^2^) [1]. Smoking status was classified into two groups: non-smokers and current smokers. Alcohol consumption was also categorised into non-drinker and current drinker groups. Sedentary time was classified into three categories: ≤2 h/day, >2–≤4 h/day, and >4 h/day.

#### 2.3.4. Health Conditions

Health conditions assessed in this study included diabetes, which was identified based on a finger-prick blood glucose measurement, with a fasting level of 7.0 mmol/L or higher or a non-fasting level of 11.1 mmol/L or higher [14] or self-reported medically diagnosed diabetes. For hypertension, blood pressure was measured twice at ten-minute intervals using a calibrated digital sphygmomanometer (OMRON HEM-907, Kyoto, Japan) during the survey. Hypertension was defined as a systolic blood pressure of 140 mmHg or higher and/or a diastolic blood pressure of 90 mmHg or higher [15], or self-reported medically diagnosed hypertension. Hypercholesterolemia was assessed using the validated CardioCheck^®^ PA device (PTS Diagnostics, Whitestown, IN, USA) to measure fasting total cholesterol from a finger-prick blood sample. Respondents were classified as having hypercholesterolemia if their cholesterol level was ≥5.2 mmol/L [16], or self-reported medically diagnosed hypercholesterolemia. Disability status was assessed using the short version of the Washington Group Questionnaire [17], which identifies functional limitations and health conditions that interfere with daily activities.

### 2.4. Data Collection Procedure

Data collection was conducted from 11 July to 29 September 2023 through door-to-door visits. Trained research assistants performed face-to-face interviews with eligible respondents in selected households, using the structured questionnaire on mobile devices.

### 2.5. Data Analysis

Data analyses were performed using SPSS version 20 (IBM SPSS Statistics for Windows, version 20 (IBM Corp., Armonk, NY, USA), considering the sample weighting and complex sampling design. Descriptive statistics, including frequencies and percentages, were used to summarise physical inactivity patterns of the study population. Multiple logistic regression was employed to identify factors associated with physical inactivity. Variables with a *p*-value of less than 0.25 in the univariate analysis were selected for inclusion in the multivariate model. All analyses, including sensitivity analyses, were conducted using a 95% confidence interval (CI).

## 3. Results

Of the 10,858 respondents, 3326 (29.9%) were classified as physically inactive. Key factors associated with higher inactivity included urban residence, female sex, older age, Indian ethnicity, being single/divorced/widowed, unemployment, and both low and high household income. Higher sedentary time (38.7%), chronic conditions (diabetes, hypertension, hypercholesterolemia), and disabilities were also significantly associated with inactivity (*p* < 0.001) (Table 1).

Bivariate analysis showed higher odds of inactivity among urban residents, females, older adults (≥60 years), those of Indian ethnicity, the unmarried, unemployed, and those reporting longer sedentary time. Chronic conditions such as diabetes, hypertension, and disabilities were also linked to inactivity. In multivariate analysis, urban residence, being single/divorced/widowed, unemployment, sedentary time >4 h/day, and diabetes remained significant predictors. Notably, age, income, and hypertension lost significance after adjustment (Table 2).

## 4. Discussion

The study found that physical inactivity is influenced by socio-demographic, lifestyle, and health factors. Urban residents showed particularly low activity levels, likely due to environmental barriers such as overcrowding, lack of recreational spaces, poor walking infrastructure, and the fast-paced nature of city life may further discourage physical activity [18]. Previous studies have consistently shown that urbanisation contributes to reduced physical activity levels through multiple factors, including traffic congestion, safety concerns, and increasingly sedentary occupations [18,19]. Compounding these issues, urban residents frequently face higher stress levels and longer working hours, which further limit opportunities for physical activity [20]. The National Health and Morbidity Survey (NHMS) conducted in 2019 found that urban residents had higher physical inactivity rates than rural populations, further highlighting how urban environments may discourage active lifestyles [8]. In Singapore, the Park Connector Network serves as an example of how infrastructure that connects parks and residential areas through cycling and walking paths can promote routine physical activity [21]. While in China, the “15 min fitness zones” policy ensures that urban residents have convenient access to recreational and sports facilities within walking distance of their homes [22].

Marital status and employment status also significantly influenced physical inactivity levels. Individuals who were single/divorced/widowed demonstrated higher inactivity rates, potentially due to diminished social support or reduced motivation for physical activity [23]. This finding aligns with existing research emphasizing the importance of social networks in promoting active lifestyles, particularly through spousal support in marital relationships [23]. Strong social support networks may help mitigate psychological barriers to physical activity, particularly for single/divorced/widowed individuals who often experience greater emotional and logistical challenges. Similarly, unemployed individuals demonstrated significantly higher inactivity levels, likely due to financial constraints and the absence of structured daily routines [24].

A local study found that individuals in the low-income group faced significant barriers to physical activity, including limited access to recreational facilities and fewer opportunities for structured exercise [25]. Poverty due to multiple jobs and caregiving responsibilities restricted opportunities for physical activity participation; in addition, the barriers were compounded by financial constraints that made even basic exercise equipment unaffordable for many households [26].

The inverse relationship between smoking and physical inactivity should be interpreted with caution, as it may reflect biases or unmeasured confounding factors rather than a true protective effect. Some evidence suggests smokers may participate in non-traditional physical activities not typically measured by standard assessment tools [27]. However, the substantial health risks associated with smoking far outweigh any incidental activity [28]. Smokers generally experience poorer health outcomes due to interrelated factors, including suboptimal nutrition, alcohol consumption, and mental health challenges [29]. Notably, even physically active smokers face significant health consequences attributable to nicotine dependence [30] and increased susceptibility to chronic conditions, particularly cardiovascular diseases [31]. These findings highlight the need for integrated public health approaches that simultaneously address smoking cessation and physical activity promotion, as combined interventions have demonstrated superior effectiveness in improving population health outcomes [32].

Overall, alcohol appears to have minimal influence on physical activity patterns in the Malaysian context. No significant association between alcohol consumption and physical inactivity was observed. This may reflect the low prevalence of drinkers in Malaysia, largely due to cultural, religious, and social norms, particularly among the Malay majority, which may limit statistical power [33]. Previous research indicates mixed findings, with moderate alcohol use sometimes associated with higher participation in social or recreational activities, while heavy drinking is linked to reduced activity [34].

Health-related factors, particularly diabetes and disabilities, emerged as strong predictors of physical inactivity. This aligns with existing evidence demonstrating that chronic conditions often create activity barriers through fatigue, pain, and mobility limitations [35,36]. The Malaysia NHMS 2015 documented increased sedentary behaviour among individuals with non-communicable diseases like diabetes, with pain and fatigue being predominant contributing factors [7]. Disability populations face particularly significant challenges, including inaccessible exercise facilities and insufficient adaptive programming [37]. Individuals with disabilities frequently encounter unsupportive infrastructure and limited resources for physical activity participation [38]. The WHO has emphasised the need for inclusive physical activity interventions tailored to individuals with chronic conditions and disabilities [39]. In Malaysia, the Ministry of Women, Family, and Community Development established the Community-Based Rehabilitation (CBR) Programme, which seeks to improve access to rehabilitation, education, and physical activity opportunities for persons with disabilities within community settings [40].

Bivariate analysis showed that sex, age, income, hypertension, and BMI were significantly associated with physical inactivity. However, these associations lost significance in the multivariate model, suggesting that their effects were confounded by other factors [9,10]. Independent predictors of inactivity included urban residence, being single/divorced/widowed, not working, prolonged sedentary time, diabetes, and disability [8,38]. These findings suggest that environmental, social, and health-related factors are plausible contributors to physical inactivity in Malaysian adults.

There are several limitations to this study. First, its cross-sectional design limits the ability to establish causal relationships and assess long-term effects, making it necessary for future longitudinal studies to validate these findings. Second, reliance on self-reported data introduces the possibility of recall bias, which could affect the accuracy of physical activity measurements. Third, data collection between July and September may have influenced activity levels, as high temperatures during this period could discourage outdoor exercise. Despite these limitations, this study has notable strengths. The use of a nationally representative sample enhances the generalizability of the findings, reducing the risk of selection bias. Furthermore, employing validated questionnaires and structured interviews strengthens the reliability and consistency of the data. Future studies should incorporate objective tools, such as accelerometers or wearable devices, to provide more precise and unbiased measurements of physical activity.

## 5. Conclusions

The prevalence of physical inactivity among adults in Malaysia remains high, particularly among urban residents, unemployed individuals, increased sedentary behaviour, and those with chronic health conditions. To address this, policymakers should prioritise: (1) urban planning strategies that promote active living, such as safe walking and cycling infrastructure; (2) workplace and community-based wellness initiatives targeting high-risk populations; and (3) inclusive fitness programs and public health campaigns that encourage physical activity across all age groups. Future studies could use moderation or mediation analyses to clarify mechanisms linking socio-demographic and health factors with inactivity. Longitudinal research is needed to assess causal relationships and trends.

## Figures and Tables

**Table 1 healthcare-13-02978-t001:** Prevalence of physical activity among Malaysian adults according to socio-demographic characteristics, lifestyle-related factors, and health conditions.

Characteristics	n	Physical Activity		*p*
		Inactive n (%)	Active n (%)	
**Overall**	**10** **,** **858**	**3326 (29.9)**	**7532 (70.1)**	
Socio-demographic characteristic				<0.001
Location				
Urban	8294	2642 (31.7)	5646 (68.3)	
Rural	2564	684 (23.6)	1880 (76.4)	
Sex				<0.001
Male	5000	1424 (27.2)	3573 (72.8)	
Female	5858	1902 (32.9)	3953 (67.1)	
Age				<0.001
18–39	4166	1121 (27.3)	3041 (72.7)	
40–49	1907	436 (25.1)	1471 (74.9)	
59	1806	506 (30.3)	1298 (69.7)	
60 and above	2979	1263 (43.2)	1716 (56.8)	
Ethnicity	6323			<0.001
Malay	1689	2096 (31.6)	4226 (68.4)	
Chinese	701	638 (36.1)	1049 (63.9)	
Indians	1043	250 (37.8)	450 (62.2)	
Other Bumiputera	412	236 (18.7)	1219 (81.3)	
Others	690	106 (14.3)	582 (85.7)	
Education				0.506
None	1091	409 (30.2)	682 (69.8)	
Primary	1435	480 (30.1)	955 (69.9)	
Secondary	6923	1994 (29.3)	4926 (70.7)	
Tertiary	1373	426 (32.4)	945 (67.6)	
Marital				<0.001
Single/divorced/widowed	3594	1223 (32.7)	2371 (67.3)	
Married	7249	2100 (28.3)	5149 (71.7)	
Occupation				<0.001
Government	994	264 (24.9)	730 (75.1)	
Private	3191	751 (25.5)	2436 (74.5)	
Self-employed	1642	317 (19.4)	1325 (80.6)	
Not working	4984	1970 (39.1)	3012 (60.9)	
Household income				<0.001
Q1 (Poorer 20%)	2266	869 (35.8)	1397 (64.2)	
Q2	2215	582 (24.6)	1633 (75.4)	
Q3	2082	561 (26.9)	1521 (73.1)	
Q4	2223	664 (30.0)	1559 (70.0)	
Q5 (Richer 20%)	2034	638 (33.2)	1396 (66.8)	
Lifestyle-related variables				0.114
BMI Status				
Underweight	471	161 (33.7)	310 (66.3)	
Normal	3853	1095 (27.0)	2757 (73.0)	
Overweight	3464	978 (28.8)	2485 (71.2)	
Obese	2342	702 (29.9)	1640 (70.1)	
Current Smoker				<0.001
No	8882	2890 (32.3)	5991 (67.7)	
Yes	1959	431 (20.5)	1527 (79.5)	
Current drinker				0.340
No	9928	3071 (30.0)	6854 (70.0)	
Yes	823	208 (27.8)	615 (72.2)	
Sedentary time				<0.001
2 h and below	5439	1538 (27.2)	3901 (72.8)	
More than 2 h to 4 h	3586	1029 (29.2)	2557 (70.8)	
More than 4 h	1812	744 (38.7)	1068 (61.3)	
Health conditions				<0.001
Diabetes				
No	8545	2443 (28.5)	6100 (71.5)	
Yes	2308	882 (37.8)	1426 (62.2)	
Hypertension				<0.001
No	6805	1863 (27.8)	4941 (72.2)	
Yes	4048	1462 (35.1)	2585 (64.9)	
Hypercholesterolaemia				0.020
No	6499	1879 (28.9)	4619 (71.1)	
Yes	4353	1446 (32.1)	2907 (67.9)	
Disability				<0.001
No	9635	2716 (28.1)	6916 (71.9)	
Yes	1207	603 (49.8)	604 (50.2)	

**Table 2 healthcare-13-02978-t002:** Bivariate and multivariate logistic regression analysis of socio-demographic, lifestyle-related factors, and health conditions of physical inactivity.

Characteristics	Odd Ratio			
	Crude OR (95%CI)	*p*	Adjusted OR (95%CI)	*p*
Overall				
Socio-demographic				
1. Location				
Urban	1.50 (1.24, 1.82)	<0.001	1.42 (1.16, 1.75)	<0.001
Rural	reference		reference	
2. Sex				
Male	Reference		Reference	
Female	1.31 (1.16, 1.48)	<0.001	0.95 (0.82, 1.11)	0.530
3. Age				
18–39	Reference		reference	
40–49	0.89 (0.76, 1.05)	<0.001	0.92 (0.76, 1.10)	0.169
50–59	1.16 (0.98, 1.36)	0.161	1.02 (0.84, 1.23)	0.341
60 and above	2.03 (1.72, 2.39)	<0.001	1.17 (0.94, 1.47)	0.880
4. Ethnicity				
Malay	reference		reference	
Chinese	1.22 (1.00, 1.49)	0.051	1.20 (0.97, 1.49)	0.090
Indians	1.31 (1.00, 1.72)	0.046	1.31 (0.98, 1.74)	0.070
Other Bumiputera	0.50 (0.40, 0.62)	<0.001	0.53 (0.42, 0.66)	<0.001
Others	0.36 (0.26, 0.51)	<0.001	0.40 (0.28, 0.57)	<0.001
5. Education *				
None	reference			
Primary	1.00 (0.77, 1.30)	0.995		
Secondary	0.96 (0.75, 1.23)	0.745		
Tertiary	1.11 (0.83, 1.49)	0.490		
6. Marital				
Single/divorced/widowed	1.23 (1.08, 1.39)	<0.001	1.16 (1.01, 1.33)	0.038
Married	reference		reference	
7. Occupation				
Government	reference		reference	
Private	1.04 (0.81, 1.31)	0.782	1.11 (0.85, 1.45)	0.457
Self-employed	0.73 (0.56, 0.94)	<0.015	0.82 (0.61, 1.09)	0.175
Not working	1.94 (1.56, 2.41)	<0.001	1.61 (1.24, 2.09)	<0.001
8. Household income				
Q1 (Poorer 20%)	reference		reference	
Q2	0.59 (0.48, 0.71)	<0.001	0.83 (0.68, 1.01)	0.124
Q3	0.66 (0.55, 0.80)	<0.001	0.88 (0.72, 1.07)	0.062
Q4	0.77 (0.64, 0.93)	0.008	1.04 (0.85, 1.28)	0.198
Q5 (Richer 20%)	0.89 (0.71, 1.13)	0.337	1.21 (0.95, 1.54)	0.701
Lifestyle-related variables				
9. BMI Status				
Underweight	1.37 (1.02, 1.85)	0.039	1.34 (1.00, 1.80)	0.050
Normal	reference		reference	
Overweight	1.09 (0.93, 1.28)	0.275	1.07 (0.91, 1.26)	0.421
Obese	1.15 (0.98, 1.35)	0.082	1.11 (0.95, 1.30)	0.199
10. Current Smoker				
No	reference		reference	
Yes	0.54 (0.46, 0.64)	<0.001	0.77 (0.63, 0.93)	0.008
11. Current drinker *		0.34		
No	reference			
Yes	0.90 (0.72, 1.12)			
12. Sedentary time				
2 h and below	reference		reference	
>2 h to 4 h	1.10 (0.94, 1.30)	0.241	1.03 (0.87, 1.22)	0.725
>4 h	1.69 (1.40, 2.03)	<0.001	1.46 (1.20, 1.76)	<0.001
Health conditions				
13. Diabetes *				
No	reference		reference	
Yes	1.53 (1.34, 1.74)	<0.001	1.17 (1.00, 1.37)	0.047
14. Hypertension				
No	Reference		reference	
Yes	1.41 (1.24, 1.60)	<0.001	1.14 (0.98, 1.32)	0.095
15. Hypercholesterolaemia				
No	reference		reference	
Yes	1.17 (1.03, 1.33)	0.020	0.99 (0.85, 1.14)	0.849
16. Disability				
No	reference		reference	
Yes	2.53 (2.15, 2.98)	<0.001	1.67 (1.39, 2.02)	<0.001

* Variable not included in the final model.

## Data Availability

The current study datasets are available from the corresponding author upon reasonable request.

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
