# Peer review of "Prevalence and Associated Factors of Physical Inactivity Among Malaysian Adults"

_healthcare, 2025, doi:10.3390/healthcare13222978_

Round 1

Reviewer 1 Report

Comments and Suggestions for Authors

Dear authors,

Thank you for the opportunity to review your manuscript.

I appreciated the reading; however, the manuscript does not add anything new to the literature and describes a Malaysian situation that is very similar to one already described in 2019.

The consequences of inactivity and its status among populations are well-known factors.

Perhaps you can conduct additional statistical analyses (for example, moderation or mediation analyses) or explore alternative ways to interpret the results.

Best regards,

Author Response

AUTHOR RESPONSE TO REVIEWER COMMENTS

 Title: Prevalence and associated factors of physical inactivity among Malaysian adults: Findings from the National Health and Morbidity Survey 2023

 Authors' responses to the Reviewer’s comments:

-         We have made all the necessary changes point-by-point according to the reviewer's comments.

  1. Response to Reviewer 1 Comments

Open Review

(x) I would not like to sign my review report

( ) I would like to sign my review report

Quality of English Language

( ) The English could be improved to more clearly express the research.

(x) The English is fine and does not require any improvement.

Does the introduction provide sufficient background and include all relevant references?

Yes       Can be improved         Must be improved       Not applicable

( )            (x)                                         ( )                                   ( )

Is the research design appropriate?

(x)           ( )                                          ( )                                  ( )

Are the methods adequately described?

( )            (x)                                         ( )                                  ( )

Are the results clearly presented?

( )            ( )                                         (x)                                   ( )

Are the conclusions supported by the results?

(x)           ( )                                         ( )                                   ( )

Are all figures and tables clear and well-presented?

( )            ( )                                         (x)                                  ( )

Comments and Suggestions for Authors

Dear authors,

Thank you for the opportunity to review your manuscript.

Comment 1: I appreciated the reading; however, the manuscript does not add anything new to the literature and describes a Malaysian situation that is very similar to one already described in 2019.  The consequences of inactivity and its status among populations are well-known factors.

Response 1: Thank you very much for the suggestion. While the consequences of physical inactivity are well-known, our study provides updated evidence on its prevalence and socio-demographic, lifestyle, and health-related correlates among Malaysian adults using nationally representative data.

Comment 2: Perhaps you can conduct additional statistical analyses (for example, moderation or mediation analyses) or explore alternative ways to interpret the results.

Response 2: Thank you for the suggestion, regarding additional analyses, the current study focused on associations using logistic regression, but we will consider moderation or mediation analyses in future work to better understand the mechanisms behind physical inactivity in Malaysia.

Best regards,

Submission Date: 18 July 2025

Date of this review: 24 Jul 2025 14:35:41

Reviewer 2 Report

Comments and Suggestions for Authors

Dear Authors,

Thank you for your submission to Healthcare. Your manuscript addresses an important public health issue, and I commend the team for conducting a large, nationally representative study. The paper is relevant, timely, and provides useful insights for health policy and practice. Below I provide detailed feedback to further improve the manuscript:

Major Revisions

Discussion on Lifestyle Variables:

The relationship between smoking and physical inactivity is intriguing (smokers appear less inactive). Please expand this section with possible explanations and refer to existing literature. Similarly, the role of alcohol consumption warrants deeper discussion.

Bivariate vs. Multivariate Results:

Some variables (e.g., income, hypertension) were significant in the bivariate analysis but lost significance in the multivariate model. Please provide a more detailed interpretation of these differences.

Tables and Figures:

The tables are comprehensive but dense. Consider adding summary graphics (e.g., bar charts, forest plots for odds ratios) to enhance readability for the audience.

Limitations Section:

Please emphasize more strongly the limitations of self-reported measures (GPAQ), including recall and social desirability bias. Future studies should be recommended to incorporate objective tools such as accelerometers or wearable devices.

Policy Implications:

While the conclusion is strong, it would benefit from more specific and actionable recommendations for policymakers (e.g., workplace wellness programs, urban design to encourage active transport).

Minor Revisions

Please ensure consistency in reporting percentages and odds ratios (e.g., two decimal places).

Revise minor grammatical inconsistencies for clarity and flow.

Author Response

AUTHOR RESPONSE TO REVIEWER COMMENTS

Title: Prevalence and associated factors of physical inactivity among Malaysian adults: Findings from the National Health and Morbidity Survey 2023

 Authors' responses to the Reviewer’s comments: We have made all the necessary changes point-by-point according to the reviewer's comments.

  1. Response to Reviewer 2 Comments

Open Review

(x) I would not like to sign my review report

( ) I would like to sign my review report

Quality of English Language

( ) The English could be improved to more clearly express the research.

(x) The English is fine and does not require any improvement.

Does the introduction provide sufficient background and include all relevant references?

Yes       Can be improved         Must be improved       Not applicable

( )                 (x)                                     ( )                                       ( )

Is the research design appropriate?

( )                 (x)                                     ( )                                       ( )

Are the methods adequately described?

( )                 (x)                                    ( )                                        ( )

Are the results clearly presented?

( )                 (x)                                    ( )                                        ( )

Are the conclusions supported by the results?

( )                 (x)                                    ( )                                         ( )

Are all figures and tables clear and well-presented?

( )                 (x)                                    ( )                                        ( )

Comments and Suggestions for Authors

Dear Authors,

Thank you for your submission to Healthcare. Your manuscript addresses an important public health issue, and I commend the team for conducting a large, nationally representative study. The paper is relevant, timely, and provides useful insights for health policy and practice. Below I provide detailed feedback to further improve the manuscript:

Major Revisions

Comment 1: Discussion on Lifestyle Variables:

The relationship between smoking and physical inactivity is intriguing (smokers appear less inactive). Please expand this section with possible explanations and refer to existing literature. Similarly, the role of alcohol consumption warrants deeper discussion.

Response 1: Thank you very much for the suggestion. I have expanded the smoking and alcohol discussion according to your suggestion. (L277 – 298)

Comment 2: Bivariate vs. Multivariate Results:

Some variables (e.g., income, hypertension) were significant in the bivariate analysis but lost significance in the multivariate model. Please provide a more detailed interpretation of these differences.

Response 2: Thank you very much for the suggestion. I have expanded the reasons for this interpretation according to your suggestion. (L 314 - 320)

Comment 3: Tables and Figures:

The tables are comprehensive but dense. Consider adding summary graphics (e.g., bar charts, forest plots for odds ratios) to enhance readability for the audience.

Response 3: We appreciate the suggestion. The tables provide a more self-explanatory and detailed presentation, including exact values, confidence intervals, and p-values, which graphs may oversimplify. Therefore, we retained the tables to ensure precise reporting.

Comment 4: Limitations Section:

Please emphasize more strongly the limitations of self-reported measures (GPAQ), including recall and social desirability bias. Future studies should be recommended to incorporate objective tools such as accelerometers or wearable devices.

Response 4: Thank you for the suggestion. We have expanded the discussion of future studies by recommending the use of objective tools, such as accelerometers or wearable devices, to obtain more accurate measurements of physical activity. (L332 – 334)

Comment 5: Policy Implications:

While the conclusion is strong, it would benefit from more specific and actionable recommendations for policymakers (e.g., workplace wellness programs, urban design to encourage active transport).

Response 5: Thank you very much for the suggestion. I have rephrased the changes according to your suggestion. (L339 – 343)

Comment 6: Minor Revisions

Please ensure consistency in reporting percentages and odds ratios (e.g., two decimal places). Revise minor grammatical inconsistencies for clarity and flow.

Response 6: Thank you very much for the suggestion. I have made the changes according to your suggestion on decimal and grammatical inconsistencies.

Submission Date: 18 July 2025

Date of this review: 20 Aug 2025 13:07:29

Reviewer 3 Report

Comments and Suggestions for Authors

This manuscript tackles a significant public health concern and leverages the nationally representative data from the 2023 National Health and Morbidity Survey in Malaysia in a compelling manner. The introduction is logically organised, presenting a sound global, regional, and national background that is underpinned by recent and pertinent references. Methodologically, the design is sound, utilising a reflective sampling framework and standardised tools, notably the GPAQ, which have been previously validated. The statistical methodology is articulated with adequate detail, while the results are conveyed in a manner that is straightforward, aided by tables that are both exhaustive and easily decipherable.  

The discussion is tightly interwoven with the empirical results, engaging with pertinent literature to situate the observed associations within a wider scholarly conversation. The manuscript candidly acknowledges both strengths and weaknesses, highlighting the limitations that accompany cross-sectional and self-reported data. The conclusions are firmly anchored in the results and offer actionable insights for both policy formulation and practical implementation.

A few minor suggestions could strengthen the paper:

  1. Clarify interpretation of certain adjusted odds ratios in the discussion, especially where the bivariate significance does not remain in the multivariate model (e.g., age and hypertension). This will help readers understand the attenuation effects.

  2. Expand on implications for intervention design. For example, differentiating strategies for urban versus rural populations, and for individuals with disabilities.

  3. Consider briefly discussing the potential role of accelerometer-based measurement in future NHMS cycles to address self-report bias.

Author Response

AUTHOR RESPONSE TO REVIEWER COMMENTS

Title: Prevalence and associated factors of physical inactivity among Malaysian adults: Findings from the National Health and Morbidity Survey 2023

Authors' responses to the Reviewer’s comments: We have made all the necessary changes point-by-point according to the reviewer's comments.

  1. Response to Reviewer 3 Comments

Open Review

( ) I would not like to sign my review report

(x) I would like to sign my review report

Quality of English Language

( ) The English could be improved to more clearly express the research.

(x) The English is fine and does not require any improvement.

Does the introduction provide sufficient background and include all relevant references?

Yes       Can be improved         Must be improved       Not applicable

(x)                     ( )                                 ( )                                  ( )

Is the research design appropriate?

(x)                     ( )                                 ( )                                   ( )

Are the methods adequately described?

(x)                      ( )                                ( )                                   ( )

Are the results clearly presented?

(x)                      ( )                                ( )                                   ( )

Are the conclusions supported by the results?

(x)                      ( )                                ( )                                   ( )

Are all figures and tables clear and well-presented?

(x)                      ( )                                ( )                                    ( )

Comments and Suggestions for Authors

This manuscript tackles a significant public health concern and leverages the nationally representative data from the 2023 National Health and Morbidity Survey in Malaysia in a compelling manner. The introduction is logically organised, presenting a sound global, regional, and national background that is underpinned by recent and pertinent references. Methodologically, the design is sound, utilising a reflective sampling framework and standardised tools, notably the GPAQ, which have been previously validated. The statistical methodology is articulated with adequate detail, while the results are conveyed in a manner that is straightforward, aided by tables that are both exhaustive and easily decipherable. The discussion is tightly interwoven with the empirical results, engaging with pertinent literature to situate the observed associations within a wider scholarly conversation. The manuscript candidly acknowledges both strengths and weaknesses, highlighting the limitations that accompany cross-sectional and self-reported data. The conclusions are firmly anchored in the results and offer actionable insights for both policy formulation and practical implementation.

A few minor suggestions could strengthen the paper:

Comment 1: Clarify interpretation of certain adjusted odds ratios in the discussion, especially where the bivariate significance does not remain in the multivariate model (e.g., age and hypertension). This will help readers understand the attenuation effects.

Response 1: Thank you very much for the suggestion. I have expanded the reasons for this interpretation according to your suggestion. (L314 - 320)

Comment 2: Expand on implications for intervention design. For example, differentiating strategies for urban versus rural populations, and for individuals with disabilities.

Response 2: Thank you very much for the suggestion. I have revised according to your suggestion. (L339 - 343)

Comment 3: Consider briefly discussing the potential role of accelerometer-based measurement in future NHMS cycles to address self-report bias.

Response 3: Thank you for the suggestion. We have expanded the discussion of future studies by recommending the use of objective tools, such as accelerometers or wearable devices, to obtain more accurate measurements of physical activity. (L347 - 349)

Submission Date: 18 July 2025

Date of this review: 11 Aug 2025 08:45:18

Reviewer 4 Report

Comments and Suggestions for Authors

Prevalence and associated factors of physical inactivity among Malaysian adults: Findings from the National Health and Morbidity Survey 2023

Aspects for improvement:

1. In the Materials and Methods section, it is suggested to include a subsection on the design and type of research. In 2.2. Instruments and Procedures, it is suggested to include a description of the instrument and the dimensions evaluated, as well as its reliability.
Regarding this section, it is important to maintain a logical sequence, such as:
Research design and type
Population and sample
Instrument technique
Data collection procedure
Data analysis
2. Results: Provide a detailed description of the tables to allow the reader a better understanding.
3. Include current studies related to the object of study in the discussion. It would be important to include two subsections: Limitations of the study and New lines of research.
4. Review the conclusions in relation to the objective of the study and the results obtained, ensuring there is harmony between them.

Author Response

AUTHOR RESPONSE TO REVIEWER COMMENTS

Title: Prevalence and associated factors of physical inactivity among Malaysian adults: Findings from the National Health and Morbidity Survey 2023

 Authors' responses to the Reviewer’s comments:  We have made all the necessary changes point-by-point according to the reviewer's comments.

  1. Response to Reviewer 4 Comments

Open Review

( ) I would not like to sign my review report

(x) I would like to sign my review report

Quality of English Language

( ) The English could be improved to more clearly express the research.

(x) The English is fine and does not require any improvement.

Does the introduction provide sufficient background and include all relevant references?

Yes       Can be improved         Must be improved       Not applicable

(x)                  ( )                                        ( )                                ( )

Is the research design appropriate?

( )                  (x)                                        ( )                                ( )

Are the methods adequately described?

( )                    ( )                                       (x)                               ( )

Are the results clearly presented?

( )                   (x)                                       ( )                                ( )

Are the conclusions supported by the results?

( )                    ( )                                       (x)                                ( )

Are all figures and tables clear and well-presented?

( )                   (x)                                       ( )                                 ( )

Comments and Suggestions for Authors

Prevalence and associated factors of physical inactivity among Malaysian adults: Findings from the National Health and Morbidity Survey 2023

Aspects for improvement:

Comment 1: In the Materials and Methods section, it is suggested to include a subsection on the design and type of research. In 2.2. Instruments and Procedures, it is suggested to include a description of the instrument and the dimensions evaluated, as well as its reliability.

Regarding this section, it is important to maintain a logical sequence, such as:

Research design and type

Population and sample

Instrument technique

Data collection procedure

Data analysis

Response 1: Thank you very much for the suggestion. I have changed the format according to your suggestion.(L87 - 179)

Comment 2: Results: Provide a detailed description of the tables to allow the reader a better understanding.

Response 2: Thank you very much for the suggestion. I have rephrased the Table 1 and table 2 according to your suggestion. (L204-205, L234-235)

Comment 3: Include current studies related to the object of study in the discussion. It would be important to include two subsections: Limitations of the study and new lines of research.

Response 3: Thank you for the suggestion. We have incorporated two recent studies related to the study topic as recommended (L253 - 258). In addition, a subsection on future research directions has been included in the conclusion section. (L347 - 349)

Comment 4: Review the conclusions in relation to the objective of the study and the results obtained, ensuring there is harmony between them.

Response 4: Thank you for the suggestion. We have rephrased the conclusion as recommended. (L337 – 343)

Submission Date: 18 July 2025

Date of this review: 08 Aug 2025 12:00:59

Reviewer 5 Report

Comments and Suggestions for Authors

Dear author:

The manuscript presents a relevant and current topic for public health in Malaysia, addressing the prevalence and factors associated with physical inactivity with nationally representative data. The overall structure is solid, the methods are appropriate, and the results are clearly presented. However, I have identified some aspects that could improve the clarity, accuracy, and impact of the article, particularly in terms of wording, justification of certain methodological points, and presentation of results.

Below, I include line-by-line comments so that you can specifically address the areas that I consider could be improved:

Line-by-line comments
Title and abstract

L1–4: The title is clear and reflects the content, but it could be shortened by removing ‘Findings from the National Health and Morbidity Survey 2023’ or moving it to the subtitle for greater conciseness.

L19–42: The abstract is informative, but I suggest indicating the cross-sectional design at the beginning to quickly orient the reader.

L37–40: The conclusion of the abstract could be strengthened with a final sentence that directly connects to public policy implications or specific recommendations.

Introduction

L44–48: Correct “The Physical inactivity…” by removing the initial “The”.

L57–64: Good summary of global prevalence, but add a more recent reference (≥2023) on global trends in physical inactivity.

L70–79: Briefly expand on why inactivity decreased between NHMS 2015 and 2019 (e.g., methodological changes, national campaigns).

Materials and methods

L82–95: Include non-response rate and how it was handled in the analysis, even though sample weighting is mentioned later.

L103–111: Specify whether the classification of inactivity (<600 MET-min/week) considers all physical activity or only leisure-time activity.

L114–119: Justify the age categories and whether they are aligned with previous NHMS studies.

L121–132: Justify the sedentary time cut-off points.

L150–156: Indicate whether sensitivity analyses were performed to assess the robustness of the results.

Results

L162–170: Avoid redundancy between text and Table 1; summarise key findings and refer the reader to the table.

L175–194: In the text, highlight only the most relevant or unexpected associations to avoid data overload.

Discussion

L200–214: Reinforce analysis of the urban environment with examples of successful policies in other Asian countries.

L231–240: Treat the inverse association between smoking and inactivity with more caution, emphasising that it could be due to biases or unmeasured factors.

L243–252: Add examples of existing inclusive programmes in Malaysia for people with disabilities, if any.

L254–259: Expand limitations on the possible influence of seasonality (July–September) on physical activity measurement.

Conclusions

L267–274: Conclude with 2–3 priority actions for policymakers.

References

Review and update key references to more recent versions; add DOI where missing (e.g. refs 7, 8, 11). Already included in the text. Have a minimum of 3 years' experience and hold a university degree or similar qualification (e.g. CSCS).

These revisions will help improve the article.

Author Response

AUTHOR RESPONSE TO REVIEWER COMMENTS

Title: Prevalence and associated factors of physical inactivity among Malaysian adults: Findings from the National Health and Morbidity Survey 2023

 Authors' responses to the Reviewer’s comments:  We have made all the necessary changes point-by-point according to the reviewer's comments

  1. Response to Reviewer 5 Comments

Open Review

(x) I would not like to sign my review report

( ) I would like to sign my review report

Quality of English Language

( ) The English could be improved to more clearly express the research.

(x) The English is fine and does not require any improvement.

Does the introduction provide sufficient background and include all relevant references?

Yes       Can be improved         Must be improved       Not applicable

( )                  (x)                                  ( )                                   ( )

Is the research design appropriate?

(x)                 ( )                                    ( )                                   ( )

Are the methods adequately described?

(x)                 ( )                                    ( )                                   ( )

Are the results clearly presented?

(x)                 ( )                                    ( )                                   ( )

Are the conclusions supported by the results?

(x)                 ( )                                   ( )                                    ( )

Are all figures and tables clear and well-presented?

(x)                 ( )                                  ( )                                     ( )

Comments and Suggestions for Authors

Dear author:

The manuscript presents a relevant and current topic for public health in Malaysia, addressing the prevalence and factors associated with physical inactivity with nationally representative data. The overall structure is solid, the methods are appropriate, and the results are clearly presented. However, I have identified some aspects that could improve the clarity, accuracy, and impact of the article, particularly in terms of wording, justification of certain methodological points, and presentation of results.

Below, I include line-by-line comments so that you can specifically address the areas that I consider could be improved:

Line-by-line comments

Title and abstract

Comment 1: L1–4: The title is clear and reflects the content, but it could be shortened by removing ‘Findings from the National Health and Morbidity Survey 2023’ or moving it to the subtitle for greater conciseness.

Response 1: Thank you very much for the suggestion. I have revised according to your suggestion. (L3 – 4)

Comment 2: L19–42: The abstract is informative, but I suggest indicating the cross-sectional design at the beginning to quickly orient the reader.

Response 2: We thank the reviewer for the comment. We have moved the cross-sectional word from L26 (method) to L23 (introduction). (L23)

Comment 3: L37–40: The conclusion of the abstract could be strengthened with a final sentence that directly connects to public policy implications or specific recommendations.

Response 3: We thank the reviewer for the comment. We have revised the sentences according to the suggestion. (L41 -43)

Introduction

Comment 4: L44–48: Correct “The Physical inactivity…” by removing the initial “The”.

Response 4: Thanks. It has been deleted. (L48)

Comment 5: L57–64: Good summary of global prevalence, but add a more recent reference (≥2023) on global trends in physical inactivity.

Response 5: Thank you very much for the suggestion. I have changed the reference according to your suggestion. (L449 - 452)

Comment 6: L70–79: Briefly expand on why inactivity decreased between NHMS 2015 and 2019 (e.g., methodological changes, national campaigns).

Response 6: Thank you very much for the suggestion. I have made a plausible reason according to your suggestion. (L79 – 82)

Materials and methods

Comment 7: L82–95: Include non-response rate and how it was handled in the analysis, even though sample weighting is mentioned later.

Response 7: Thank you for the suggestion. I have added a statement according to your suggestion. (L98 - 99)

Comment 8: L103–111: Specify whether the classification of inactivity (<600 MET-min/week) considers all physical activity or only leisure-time activity.

Response 8: Thank you for the comment. The sentence has been improved after the change. (L123 – 126)

Comment 9: L114–119: Justify the age categories and whether they are aligned with previous NHMS studies.

Response 9: Thank you for the suggestion. I have added a statement according to your suggestion. (L135 – 136)

Comment 10: L121–132: Justify the sedentary time cut-off points.

Response 10: Thank you for the suggestion. I have added a new statement according to your suggestion. (L126 – 127)

Comment 11: L150–156: Indicate whether sensitivity analyses were performed to assess the robustness of the results.

Response 11: Thank you for the suggestion. Sensitivity analyses were performed in this study, and I have added a statement in the manuscript. (L181)

Results

Comment 12: L162–170: Avoid redundancy between text and Table 1; summarise key findings and refer the reader to the table.

Response12: Thank you very much for the suggestion. I have made the changes according to your suggestion. (L192 – 197, L212 – 218)

Comment 13: L175–194: In the text, highlight only the most relevant or unexpected associations to avoid data overload.

Response 13: Thank you very much for the suggestion. I have made the changes according to your suggestion. (L212 – 218)

Discussion

Comment 14: L200–214: Reinforce analysis of the urban environment with examples of successful policies in other Asian countries.

Response 14: Thank you very much for the suggestion. I have added new statements and references  according to your suggestion. (L255-259)

Comment 15: L231–240: Treat the inverse association between smoking and inactivity with more caution, emphasising that it could be due to biases or unmeasured factors.

Response 15: Thank you for the suggestion. I have added a statement according to your suggestion. (L278-284)

Comment 16: L243–252: Add examples of existing inclusive programmes in Malaysia for people with disabilities, if any.

Response 16: Thank you for the suggestion. I have added a statement and reference according to your suggestion. (L310 – 313)

Comment 17: L254–259: Expand limitations on the possible influence of seasonality (July–September) on physical activity measurement.

Response 17: Thank you very much for the suggestion. I have added third limitation according to your suggestion. (L325 - 327)

Conclusions

Comment 18: L267–274: Conclude with 2–3 priority actions for policymakers.

Response 18: Thank you very much for the suggestion. I have revised the manuscript in line with your suggestion. (L339 – 343)

References

Comment 19: Review and update key references to more recent versions; add DOI where missing (e.g. refs 7, 8, 11).

Response 19: Thanks for pointing out our mistakes. We have revised accordingly. (L376 – 465)

These revisions will help improve the article.

Submission Date: 18 July 2025

Date of this review: 10 Aug 2025 12:02:05

Round 2

Reviewer 3 Report

Comments and Suggestions for Authors

The 2023 National Health and Morbidity Survey was used in this manuscript to examine physical inactivity and its implications among Malaysian adults. This analysis, while timely, also has relevance to policy. This research, on the whole, was mapped out and executed with methodological soundness. This was done through the use of a nationally representative sample, a validated GPQA instrument, and the appropriate statistical techniques. A discussion on the results was presented, and while comprehensive, the accompanying tables were also well elucidated to provide clarity. This is all done on considering the international relations of the subject. This manuscript is further aided by strong methodological transparency, the wide scope of the dataset, and relevant limitations.

Touching on the public health aspect of this manuscript, we can see how it associates the issues of physical inactivity and the demographic, lifestyle, and health-related factors that correlate with the problem and the populations that need the most attention. The policy implications underline the recommendations for urban planning, workplace and community approaches, and comprehensive programming. The revision of this manuscript concerning the use of odds ratios in the interpretation and the implications of the intervention measures proposed for the suggesting future objective measurements has greatly improved the manuscript.

This is a strong and urgent manuscript and a relevant piece that is scientifically accurate and well structured. It is of great interest to the Healthcare audience. For it to be publishable, the marked minor edits need to be done.

Author Response

Comments and Suggestions for Authors

The 2023 National Health and Morbidity Survey was used in this manuscript to examine physical inactivity and its implications among Malaysian adults. This analysis, while timely, also has relevance to policy. This research, on the whole, was mapped out and executed with methodological soundness. This was done through the use of a nationally representative sample, a validated GPQA instrument, and the appropriate statistical techniques.

Response 1: Thank you very much for the comments.

A discussion on the results was presented, and while comprehensive, the accompanying tables were also well elucidated to provide clarity. This is all done on considering the international relations of the subject. This manuscript is further aided by strong methodological transparency, the wide scope of the dataset, and relevant limitations.

Response 2: Thank you very much for your valuable comments. Furthermore, we have refined the table 1 and Table 2 to enhance its clarity and comprehensiveness.

Touching on the public health aspect of this manuscript, we can see how it associates the issues of physical inactivity and the demographic, lifestyle, and health-related factors that correlate with the problem and the populations that need the most attention. The policy implications underline the recommendations for urban planning, workplace and community approaches, and comprehensive programming. The revision of this manuscript concerning the use of odds ratios in the interpretation and the implications of the intervention measures proposed for the suggesting future objective measurements has greatly improved the manuscript.

Response 3: We thank the reviewer for the comments and suggestions.

This is a strong and urgent manuscript and a relevant piece that is scientifically accurate and well structured. It is of great interest to the Healthcare audience. For it to be publishable, the marked minor edits need to be done.

Response 4: We thank the reviewer for the comment. All suggested minor revisions have been carefully addressed to further enhance the quality of the manuscript.

Reviewer 4 Report

Comments and Suggestions for Authors

PREVALENCE AND ASSOCIATED FACTORS OF PHYSICAL INACTIVITY AMONG MALAYSIAN ADULTS: FINDINGS FROM THE NATIONAL HEALTH AND MORBIDITY SURVEY 2023

The authors have incorporated the suggestions made in the first review, improving the quality of the article.

Author Response

Comments and Suggestions for Authors

PREVALENCE AND ASSOCIATED FACTORS OF PHYSICAL INACTIVITY AMONG MALAYSIAN ADULTS: FINDINGS FROM THE NATIONAL HEALTH AND MORBIDITY SURVEY 2023

The authors have incorporated the suggestions made in the first review, improving the quality of the article.

Response 1: Thank you very much for your valuable comments. Furthermore, we have refined the table 1 and Table 2 to enhance its clarity and comprehensiveness.

Reviewer 5 Report

Comments and Suggestions for Authors

The manuscript has been substantially improved by the proposed changes. Congratulations.

Author Response

Comments and Suggestions for Authors

The manuscript has been substantially improved by the proposed changes. Congratulations.

Response 1: We thank the reviewer for the comment. Furthermore, we have refined the table 1 and Table 2 to enhance its clarity and comprehensiveness.

Round 3

Reviewer 3 Report

Comments and Suggestions for Authors

The authors have done a good job of responding to reviewer feedback. The manuscript is now stronger. The Discussion, in particular, provides more relevant details.